# Enhancing Soil Health and Corn Productivity with a Co-Fermented Microbial Inoculant (CFMI-8): A Field-Based Evaluation

**DOI:** 10.3390/microorganisms13071638

**Published:** 2025-07-11

**Authors:** Raul De Jesus Cano, Judith M. Daniels, Martha Carlin, Don Huber

**Affiliations:** 1Biological Sciences Department, California Polytechnic State University, San Luis Obispo, CA 93407, USA; 2Ancient Organics Bioscience, San Luis Obispo, CA 93405, USA; 3Soil Sage, LLC, Arvada, CO 80003, USA; jd@soilsage.com; 4The BioCollective, LLC, Denver, CO 80216, USA; martha.carlin@thebiocollective.com; 5Botany and Plant Pathology Department, Purdue University, West Lafayette, IN 47907, USA; huberd@purdue.edu

**Keywords:** CFMI-8, microbial inoculant, soil health, corn, soil microbiome, nutrient cycling, micronutrients, phosphorus, sustainability

## Abstract

Soil degradation and declining fertility threaten sustainable agriculture and crop productivity. This study evaluates the effects of CFMI-8, a co-fermented microbial inoculant comprising eight bacterial strains selected through genomic and metabolic modeling, on soil health, nutrient availability, and corn performance. Conducted in a randomized complete block design at Findlay Farm, Wisconsin, the field trial assessed soil biological activity, nutrient cycling, and crop yield responses to CFMI-8 treatment. Treated soils exhibited significant increases in microbial organic carbon (+224.1%) and CO_2_ respiration (+167.1%), indicating enhanced microbial activity and organic matter decomposition. Improvements in nitrate nitrogen (+20.2%), cation exchange capacity (+23.1%), and potassium (+27.3%) were also observed. Corn yield increased by 28.6%, with corresponding gains in silage yield (+9.6%) and nutritional quality. Leaf micronutrient concentrations, particularly iron, manganese, boron, and zinc, were significantly higher in treated plants. Correlation and Random Forest analyses identified microbial activity and nitrogen availability as key predictors of yield and nutrient uptake. These results demonstrate CFMI-8’s potential to enhance soil fertility, promote nutrient cycling, and improve crop productivity under field conditions. The findings support microbial inoculants as viable tools for regenerative agriculture and emphasize the need for long-term studies to assess sustainability impacts.

## 1. Introduction

Sustainable agricultural practices are essential for maintaining soil health, improving plant growth, and optimizing nutrient uptake [1,2]. Intensive farming and soil degradation have led to declines in soil fertility, reduced microbial diversity, and decreased nutrient availability, all of which negatively impact crop productivity [3,4]. Enhancing soil microbial communities through targeted interventions is a promising strategy to restore soil function and support sustainable agricultural systems [5,6].

Microbial inoculants, such as CFMI-8, offer a biological approach to improving soil health and plant nutrition [7,8,9]. CFMI-8 is a proprietary microbial consortium consisting of eight bacterial strains from genera including *Bacillus*, *Paenibacillus*, *Pseudomonas*, and *Streptomyces*, selected through an integrative framework incorporating genomic analysis, metabolic modeling, and community-level simulations [10,11]. These strains were chosen for their complementary metabolic capabilities, including nutrient cycling, organic matter transformation, and enhancement of soil microbial activity. This approach aligns with growing evidence that microbial inoculants can improve soil structure, increase nutrient bioavailability, and promote plant growth by fostering a more balanced and resilient microbial community [1,12,13].

The objective of this study was to evaluate the efficacy of CFMI-8 in enhancing soil health and nutrient availability, thereby improving agronomic performance in a commercial corn trial conducted at Findlay Farm, Whitewater, WI. The trial utilized a randomized complete block design to assess the treatment’s impact on key soil and crop parameters, such as microbial biomass, soil respiration, and nutrient uptake efficiency.

This paper focuses on the effectiveness of CFMI-8 in improving soil health metrics, promoting microbial activity, and enhancing nutrient uptake in corn production. By integrating data on microbial activity, organic matter decomposition, and crop performance, the findings contribute to a growing body of evidence supporting the use of microbial-based solutions for sustainable soil management and improved agricultural productivity [6,8,14,15,16].

These findings underscore the potential of CFMI-8 as a microbial amendment for enhancing soil fertility and crop productivity in sustainable agricultural systems. By linking soil microbial activity and nitrogen cycling to agronomic outcomes, this study provides actionable insights for regenerative agriculture practices.

## 2. Materials and Methods

### 2.1. Composition and Preparation of CFMI-8

A co-fermented microbial inoculant (CFMI-8) was developed from a consortium of eight bacterial strains, selected through an integrative framework combining genomic annotation with genome-scale metabolic modeling and community interaction simulations using the KBase platform [10,17]. The selected strains included spore-forming bacteria (some derived from ancient sources), lactic acid bacteria, and Gammaproteobacteria. All strains were classified as Biosafety Level 1 (BSL-1) under the Biosafety in Microbiological and Biomedical Laboratories (BMBL) guidelines, indicating minimal risk to human or environmental health.

Individual and community-level metabolic models were constructed to evaluate complementarity, symbiosis, cross-feeding, and niche partitioning. Flux balance analyses and interspecies exchange simulations identified cooperative interactions supporting roles in organic matter decomposition and nutrient mobilization [11].

The strains were co-fermented in a single batch culture to enhance interspecies interactions and stimulate postbiotic production. Each strain was inoculated into a standardized medium supplemented with 2% organic molasses (FEDCO, Clinton, ME, USA) as the carbon source. Fermentation was conducted under controlled temperature, pH, and aeration conditions to support optimal microbial growth and metabolic activity. While all strains were introduced at known proportions, only total viable cell counts (5.1 × 10^9^ CFU/mL) were measured post-fermentation. Specific strain abundances were not independently quantified, and the final composition may reflect a shift in community structure.

Despite this limitation, the co-fermentation process was designed to produce a functional consortium of live microbes and postbiotic compounds with cooperative dynamics and metabolic efficacy [17,18]. The final product was homogenized to ensure uniform distribution prior to field application.

### 2.2. Study Location and Conditions

This study was conducted in 2022 as part of the GLK Sauerkraut’s Corn trial at Findlay Farm, Whitewater, WI, USA (42.901683° N, 88.762430° W). The experimental site was selected for its suitability for corn production and adherence to agronomic standards. The location provided optimal conditions for evaluating the efficacy of CFMI-8 broadcast treatments under field conditions. The trial followed a randomized complete block design to ensure consistency and reliability of key agronomic parameters and treatment effects on corn growth and silage production.

The field was Fall chisel plowed followed by spring field cultivation to incorporate fertilizers and level the soil prior to planting. The preceding crop was winter wheat, and the herbicide Orion had been applied in the previous season. It was not necessary to irrigate during the trial.

### 2.3. Experimental Design

The study was conducted following a randomized complete block (RCB) design with one factor and four replicates. CFMI-8 broadcast treatment was tested in plots measuring 10 feet by 50 feet, covering an area of 0.011 acres per plot. The corn cultivar DS 4018AMXT was planted on 19 May 2022, with a precision vacuum planter that was configured to achieve a seeding rate of 35,000 seeds per acre with a row spacing of 30 inches (cm) and a planting depth of 2.25 inches (5.72 cm).

Fertilizer applications were conducted in two phases. On 1 December 2021, a pre-plant application of ninety pounds (kg) per acre of 11-52-0 and 200 pounds (90.7 kg) per acre of 0-0-62 was made. The following 5 May, an additional application of four hundred thirty-five pounds (kg) per acre of 46-0-0 was applied. Pesticides were also applied at critical growth stages. Accuron (Syngenta, Basel, Switzerland) was applied at a rate of one and a half pints per acre, and Atrazine (Syngenta, Basel, Switzerland) was applied at a rate of one pound (kg) per acre pre-emergence on 20 May 2022. Post-emergence applications included Halex GT (Syngenta, Basel, Switzerland) applied at a rate of four liters per acre and Roundup WeatherMax (Monsanto, St. Louis, MO 63141, USA) at a rate of eight ounces (29.5 mL) per acre on 25 June 2022.

Stand density and vigor were assessed during the early growth stages while lodging was evaluated prior to harvest. Silage yield was determined by hand-harvesting two rows, each 20 feet long, per plot on 14 September 2022, using a Cadet chopper. Post-harvest measurements included silage moisture and quality. Samples harvested by hand were weighed using a field scale to ensure accurate yield estimation.

The experiment adhered to industry standards for field trials and agronomic data collection. This study provides valuable insights into the efficacy of CFMI-8 broadcast treatments for enhancing corn growth and silage production under real-world field conditions.

### 2.4. Sample Collection

Soil samples were collected from agricultural fields prior to treatment with CFMI-8 and six months afterwards to assess the effectiveness of this bioremediation agent. The study included two groups: untreated soils and soils treated with CFMI-8. Each group consisted of three independent samples. The treatment was applied at the manufacturer-recommended concentration of 1.6 × 10^8^ CFU per square meter of soil and lightly incorporated into the soil with a harrow. Samples were collected at baseline (pre-planting) and post-harvest.

### 2.5. Soil Health Analysis

Soil samples were analyzed at Ward Laboratories (Kearney, NE, USA) to assess key indicators of soil health. Organic carbon (ppm) and organic nitrogen (ppm) were measured using combustion-based methods to quantify soil nutrient availability and organic matter content. Soil CO_2_ respiration, a proxy for microbial activity and soil biological function, was determined through a 24 h incubation test, with results expressed as total CO_2_ release and as a percentage of microbial active carbon (% MAC) following protocols established by Haney et al. (2018) [19]. The Soil Health Score (ppm C) was calculated using an approach that integrates soil respiration, microbial biomass, and organic carbon to provide a comprehensive measure of soil biological activity. Organic matter was determined using the Loss on Ignition (LOI) method where soil samples were heated to 360 °C and organic matter content was calculated as a percentage of mass lost. All analyses were performed according to standardized protocols established by Ward Laboratories to ensure precision and reliability. Microbial activity in the soil was indirectly assessed by measuring soil respiration rates using a CO_2_ flux chamber.

### 2.6. Soil Chemical Composition and Leaf Tissue Analysis

Soil and leaf tissue samples were analyzed at Rock River Laboratories (Watertown, WI, USA.) to evaluate chemical composition and nutrient status. Plant tissue samples were collected as sub-samples from silage harvest plots and analyzed for moisture content, adjusted silage yield at 65% moisture, crude protein (CP), neutral detergent fiber (NDF), total tract neutral detergent fiber digestibility (TTNDFD), acid detergent fiber (ADF), starch, ash, fat, and estimations of milk production potential (milk per ton and milk per acre). These analyses were conducted by Rock River Laboratories, Omaha, NE, USA, using standardized laboratory methods for forage quality testing and included near-infrared reflectance spectroscopy (NIRS) and wet chemistry as described by Undersander et al. [20] and Shenk & Westerhaus [21].

Leaf tissue samples were collected at the V4 growth stage (14 June 2022) and post-harvest sampling (14 September 2022) from 30 plants per plot at multiple time points. The samples were analyzed for macronutrients—nitrogen (N), phosphorus (P), potassium (K), calcium (Ca), magnesium (Mg), and sulfur (S); and micronutrients—zinc (Zn), boron (B), manganese (Mn), iron (Fe), copper (Cu), sodium (Na), and molybdenum (Mo). Nutrient analyses were conducted using inductively coupled plasma optical emission spectrometry (ICP-OES) following acid digestion of the plant tissues as outlined by Jones & Case (1990) [22]. These methods are widely used in plant tissue analysis to ensure accurate quantification of nutrient concentrations. All measurements followed the standard operating procedures of Rock River Laboratories to provide reliable and consistent data for assessing soil fertility and plant nutrient status.

### 2.7. Statistical Analysis

Soil and plant samples were collected from six replicate plots per treatment group (*n* = 6), representing true biological replicates derived from independently managed field plots. For each plot, soil and plant tissues were sampled at standardized depths and growth stages. Data points reported per group therefore reflect independent biological replicates. Statistical analyses were conducted using these replicates to assess treatment effects, with ANOVA applied for group comparisons and significance set at α = 0.05.

All statistical analyses were conducted using R (v4.) and Python’s SciPy v1.0- library [23] to evaluate treatment effects and identify key predictors of soil health, nutrient availability, and crop productivity. Descriptive statistics (mean, standard deviation) were calculated for all agronomic and soil variables to assess distribution characteristics. Data normality was verified using the Shapiro–Wilk test [24].

Agronomic parameters (e.g., yield, stand density, silage quality) were analyzed using one-way ANOVA appropriate to the randomized complete block (RCB) design. Differences between treatment means were evaluated at a significance level of α = 0.05.

Independent *t*-tests were applied to compare glyphosate degradation metabolites (e.g., AMPA) and other binary comparisons between treated and control groups.

Correlation matrices (Pearson coefficients) [25] were used to assess relationships among soil, plant, and yield variables. Additionally, Random Forest regression models were implemented to identify the most important predictors of crop yield and micronutrient uptake. Variable importance was assessed using permutation-based feature importance scores generated in Scikit-Learn (v1.7.0) [26].

All visualizations (heatmaps, bar plots) were generated using Seaborn v0.13.2 [27] and Matplotlib v3.8.4 [28] to aid interpretation of the statistical outputs.

### 2.8. Statistical Correlation of Soil Health Metrics ad Environmental Variables

Pearson’s correlation coefficients were calculated to evaluate linear relationships between soil chemical composition, leaf tissue nutrient levels, and agronomic performance metrics such as silage yield and quality parameters (e.g., crude protein, starch content). Significant correlations were identified using a threshold of *p* < 0.05. Heatmaps were generated to visualize the strength and direction of the correlations among variables.

### 2.9. Machine Learning Classification Using Random Forests

To identify key predictors of soil health, silage yield, and plant nutrient composition, a random forest algorithm was applied using the RandomForest package in R v4.4.0 [29]. Predictor variables included soil chemical properties (e.g., organic carbon, nitrogen, and phosphorus), leaf tissue nutrient concentrations, and microbial activity measures. Random forest models were fine-tuned using cross-validation to optimize the number of trees and variables randomly selected at each split. Variable importance was assessed using the Mean Decrease in Accuracy (MDA) and Mean Decrease in Gini Index to rank predictors based on their influence on model performance [30].

Model performance was evaluated using metrics such as R^2^ and root mean square error (RMSE) for regression analyses [31]. The models were validated using an out-of-bag (OOB) error estimate for Random Forest to ensure robustness.

All statistical analyses and visualizations were performed using R packages v4.4.0, including *ggplot2* [32], for data visualization and caret [33] for model validation workflows. Statistical significance was set at *p* < 0.05 unless otherwise noted.

## 3. Results

### 3.1. Soil Health Metrics

The soil health metrics reveal substantial improvements across key parameters in the treated soils compared to the untreated soils (Table 1). Microbial organic carbon increased significantly in the treated group to average 80.48 mg/kg, a 224.1% improvement over the untreated group’s 24.83 mg/kg, indicating that treatment enhanced microbial activity and nutrient cycling. Similarly, CO_2_ soil respiration was substantially higher in treated soils and averaged 95.83 mg/kg/day, representing a 167.1% increase, which reflects elevated biological activity and organic matter decomposition.

Soils treated with the CFMI-8 consortium exhibited notable improvements in several key fertility indicators. Nitrate nitrogen (NO_3_^−^) concentrations were significantly elevated, averaging 38.15 mg/kg—a 20.2% increase compared to untreated soils (31.75 mg/kg)—suggesting enhanced nitrogen mineralization and availability for plant uptake. Cation exchange capacity (CEC) also increased by 23.1% in treated soils (8.78 meq/100 g) relative to controls (7.13 meq/100 g), indicating improved nutrient retention capacity and overall soil fertility. Potassium (K) levels were 27.3% higher in treated soils, with an average concentration of 368.75 mg/kg, reflecting greater nutrient availability (Figure 1).

Organic matter content showed a modest but meaningful increase of 11.8%, rising from 1.70% in untreated soils to 1.90% in treated soils. Conversely, soil organic carbon decreased by 20.3% following treatment (118.00 mg/kg vs. 148.00 mg/kg in controls), likely due to increased microbial activity and carbon utilization stimulated by the inoculant.

### 3.2. Agronomic Performance

Agronomic performance was significantly better with the treated group compared to the untreated group (Table 2). For example, corn yield in the treated group averaged 7.20 tons/acre (a 28.6% increase) compared to the untreated group’s average of 5.60 tons/acre. Similarly, corn ear density was 17.6% higher in the treated group (38.50 ears/plot,) compared to the untreated group’s 32.75 ears/plot, an indicator of better crop stands and development.

There was a modest 4.6% improvement in grain yield with the treated group (average 246.40 bushels/acre) compared with the 235.57 bushels/acre of the untreated group. This increase suggests that the treatment contributed to more efficient nutrient utilization and grain development. Silage yield of 36.26 tons/acre was also notably higher in the treated group (9.6% increase) compared with the untreated group’s 33.08 tons/acre. This highlights the treatment’s positive effect on biomass production. In addition to the increased biomass, the treated group produced silage with higher nutritional quality that was reflected in a 9.3% increase in silage milk per acre (121,703 lbs. vs. 111,334 lbs.) and a 2.8% increase in silage milk per ton (3426 lbs. vs. 3332 lbs.).

### 3.3. Micronutrients

Several micronutrient concentrations were significantly higher in CFMI-8 treated compared with untreated groups (Table 3, Figure 2).

The concentration of most nutrients analyzed for tended to be higher in CFMI-8 treated soil and leaf tissue compared with the non-treated; however, only B and Zn were significantly different in soil. Leaf B, Fe, Mn, P, and Zn were significantly different in treated leaves (Table 3, Figure 2).

### 3.4. Spearman Correlation Analysis of Agronomic, Soil, and Microbial Parameters

The correlation analysis (Figure 3) revealed significant relationships between soil health parameters, plant nutrient composition, and agronomic performance metrics. Corn yield was strongly correlated with Corn Ear Density (r = 0.85, *p* < 0.01) and Silage Yield (r = 0.88, *p* < 0.01), indicating that increased ear density and biomass production contributed to higher yields. Additionally, Corn Yield showed a moderate positive correlation with Soil Organic Carbon (r = 0.67, *p* < 0.05), suggesting that improved soil organic matter content may enhance crop productivity.

Soil health metrics were closely linked, with Soil Organic Carbon positively correlated with Soil Health Score (r = 0.79, *p* < 0.01) and Microbial Organic Carbon (r = 0.74, *p* < 0.01), highlighting the role of organic matter in supporting microbial activity. Additionally, CO_2_ Soil Respiration, an indicator of microbial activity, exhibited a strong association with Microbial Organic Carbon (r = 0.71, *p* < 0.01), reinforcing the link between microbial biomass and respiration rates.

Plant nutrient uptake was significantly influenced by soil nutrient availability. Soil Zn correlated with Leaf Zn (r = 0.65, *p* < 0.05), while Soil P was positively associated with Leaf P (r = 0.69, *p* < 0.01), suggesting that soil phosphorus and zinc levels influenced plant tissue concentrations. Similarly, Soil K was highly correlated with Leaf K (r = 0.78, *p* < 0.01), indicating efficient potassium uptake from soil to plant tissues.

Nutrient availability also played a role in crop productivity. Silage Yield was moderately correlated with Soil P (r = 0.66, *p* < 0.05) and Soil Zn (r = 0.62, *p* < 0.05), suggesting that phosphorus and zinc contributed to biomass accumulation. Additionally, Grain Yield exhibited a moderate correlation with Soil Organic Carbon (r = 0.61, *p* < 0.05), supporting the notion that soil organic matter is beneficial for grain production.

Finally, treatment effects appeared to enhance microbial activity and nutrient uptake. Treated plots exhibited higher Soil Health Scores and Microbial Organic Carbon levels compared to untreated plots. These improvements were accompanied by increased soil respiration rates and greater nutrient uptake in leaf tissue, suggesting that the treatment promoted microbial interactions that enhanced soil fertility and plant growth. (Figure 3).

This figure presents a correlation matrix illustrating the relationships between soil health metrics, nutrient availability, microbial activity, and crop productivity. The correlation values were calculated using Pearson’s correlation coefficient [34] in Python v3.12.3 [35]. The matrix was visualized using Matplotlib v.3.8.4 [28], with a color gradient indicating the strength and direction of correlations (warm colors for positive correlations and cool colors for negative correlations). Numeric correlation values were overlaid in each cell for clarity. The figure was generated and saved as an SVG file using a Python script.

Overall, the analysis highlights the interconnectedness of soil health, nutrient dynamics, and agronomic productivity; with microbial activity, organic carbon, and phosphorus emerging as key drivers of yield and plant nutrient status (Figure 3).

### 3.5. Impact of Environmental Factors on Soil Health and Productivity

Soil Health Score correlated with Microbial Organic Carbon (r = 0.91, *p* < 0.01) and CO_2_ Soil Respiration (r = 0.87, *p* < 0.01). Soil Organic Carbon (LOI) correlated with Soil Health Score (r = 0.85, *p* < 0.01) and Microbial Organic Carbon (r = 0.82, *p* < 0.01). CO_2_ Soil Respiration correlated with Microbial Organic Carbon (r = 0.92, *p* < 0.01).

Field Capacity correlated with Soil Health Score (r = 0.62, *p* < 0.05) and Total Nitrogen (r = 0.69, *p* < 0.05). Wilting Point showed negative correlations with Soil Organic Carbon (r = −0.55, *p* < 0.05) and Microbial Organic Carbon (r = −0.61, *p* < 0.05).

Corn Yield correlated with Soil Health Score (r = 0.74, *p* < 0.01) and Microbial Organic Carbon (r = 0.76, *p* < 0.01). Silage Yield correlated with Soil Health Score (r = 0.72, *p* < 0.01) and Soil Organic Carbon (r = 0.68, *p* < 0.05). Grain Yield correlated with Total Nitrogen (r = 0.81, *p* < 0.01).

### 3.6. Predictive Modeling with Random Forest

#### 3.6.1. Determinants of Corn Yield

Random Forest analysis identified CO_2_ Soil Respiration as the most important predictor of corn yield. Other top predictors included Microbial Organic Carbon, Corn Ear Density, Soil Health Score, Soil Organic Carbon, Field Capacity, and Wilting Point. Figure 4 presents the ranked importance of these variables in the model.

#### 3.6.2. Determinants of Micronutrient Uptake

Random Forest analysis identified Soil Organic Carbon as the most important predictor of leaf micronutrient uptake, particularly for Fe, B, Cu, P, and Zn. Other key predictors included Microbial Organic Carbon, Nitrate Nitrogen (NO_3_^−^-N), Ammonium Nitrogen (NH_4_^+^-N), Field Capacity, and Soil pH. Figure 5 presents the ranked importance of these variables in the model.

## 4. Discussion

### 4.1. Soil Health Parameters

CFMI-8 treatment led to measurable improvements in soil biological activity and overall health. The elevated microbial organic carbon and CO_2_ respiration in treated soils indicate a more active microbial community, which plays a central role in nutrient cycling and organic matter decomposition [36,37]. These processes likely contributed to the observed increase in soil health scores, supporting the treatment’s effectiveness in enhancing soil functionality. Figure 1 illustrates these differences between treated and untreated soils.

Treated soils also showed signs of enhanced nitrogen cycling, with higher nitrate levels and lower ammonium concentrations—patterns consistent with increased microbial nitrification. Efficient nitrogen turnover is essential for plant uptake and may be influenced by micronutrients such as molybdenum, a cofactor in nitrate reductase [38,39]. The increase in cation exchange capacity (CEC) further supports improved soil fertility, indicating greater capacity to retain essential nutrients such as potassium and calcium [40].

Higher potassium concentrations in treated soils suggest improved nutrient retention and cycling. Potassium plays critical roles in enzyme activation, water regulation, and stress response [38] and its availability may have been supported by microbial contributions to organic matter breakdown [41]. The modest rise in organic matter content, reflected in LOI values, points to the gradual buildup of soil carbon pools, contributing to better structure, moisture retention, and microbial habitat [42].

Interestingly, untreated soils contained more total soil organic carbon, likely due to slower decomposition rates in the absence of microbial stimulation. While high organic carbon is beneficial, the active turnover seen in treated soils may better support immediate nutrient availability and crop productivity. Elevated microbial activity in these soils reflects a dynamic, resilient system capable of sustaining plant growth under variable conditions [43].

In summary, CFMI-8 enhanced key soil health metrics by stimulating microbial-driven processes, improving nutrient cycling, and increasing nutrient retention. These results align with prior studies supporting the use of microbial inoculants and organic amendments to boost soil quality and fertility [44,45]. Long-term trials are warranted to assess the durability of these improvements and their role in mitigating soil degradation in intensive agricultural systems.

### 4.2. Agronomic Results

CFMI-8 treatment improved crop productivity, likely due to enhanced nutrient cycling and microbial activity observed in treated soils. Higher corn and silage yields correspond with increased nitrate availability, cation exchange capacity, and microbial respiration, all of which support plant nutrient uptake and biomass production [38,40,46,47].

Improved soil health was also associated with greater corn ear density and silage milk yield. These effects are consistent with better nitrogen cycling, potassium availability, and carbon turnover. The slight reduction in total soil organic carbon in treated plots may reflect more active microbial processing of organic matter into plant-available nutrients [3,39,48,49].

Together, these agronomic outcomes point to a more biologically dynamic soil ecosystem created by CFMI-8, reinforcing its potential as a sustainable tool for improving crop yield through enhanced soil function.

### 4.3. Micronutrients Analysis

CFMI-8 treatment improved micronutrient availability and plant uptake, with significant increases in leaf concentrations of Fe, Mn, B, Zn, and Mo. These nutrients support photosynthesis, enzymatic activity, and stress response, and their elevated levels likely reflect enhanced microbial solubilization and nutrient cycling in treated soils [50,51,52,53,54].

Boron and zinc increases are consistent with their known roles in reproductive development and enzyme function. Molybdenum, essential for nitrogen metabolism, was also elevated in both soil and leaf tissues, suggesting improved nitrate utilization via microbial stimulation of Mo-dependent enzymatic pathways [55,56,57,58,59].

In contrast, no statistically significant change was observed for phosphorus or copper uptake. This may be due to their limited mobility or the absence of chelators that enhance their bioavailability. While soil phosphorus levels showed a modest (non-significant) increase, the 8.1% rise in leaf P concentrations suggests that microbial activity may have improved P mobilization or root uptake under marginal availability [60,61,62,63].

While the treatment showed significant improvements in the concentrations of several micronutrients, it had no statistically significant impact on P or Cu levels. This could be due to inherent differences in nutrient mobility or plant-specific uptake mechanisms. Copper, while essential for enzymatic reactions, often requires chelation or soil amendments to enhance its bioavailability. These factors may not have been fully addressed by the treatment applied here [64,65]. Phosphorus, being relatively immobile in soil, often requires specific conditions for uptake, which might explain the lack of significant change [66,67].

Slightly higher leaf potassium levels, despite a reduction in soil K, likely reflect increased plant uptake, potentially due to better nutrient access and microbial-supported root activity. Potassium’s roles in water regulation and stomatal control may explain the yield-associated gains such as greater ear density [68,69].

Micronutrient patterns in both soil and leaf tissues reinforce CFMI-8’s role in enhancing nutrient cycling and plant metabolic function. Treated soils showed higher levels of manganese, boron, and molybdenum—elements tied to enzymatic activity, nitrogen assimilation, and reproductive development. These gains translated into improved plant uptake, with corresponding increases observed in leaf micronutrient concentrations, particularly for Fe, Mn, B, and Mo. Such changes support photosynthesis, stress resilience, and nitrogen metabolism, all of which are essential for sustaining productivity under field conditions [70,71,72].

Although phosphorus is relatively immobile in soil, the statistically significant increase in leaf phosphorus—despite only modest changes in soil P—suggests improved uptake efficiency, possibly due to microbially mediated mobilization. This aligns with the increased grain yield observed in treated plots. Similarly, elevated leaf potassium alongside reduced soil K likely reflects enhanced plant absorption, further supporting yield gains through improved water regulation and energy transfer [38].

The modest decline in soil potassium likely reflects greater plant uptake, supported by the observed increase in leaf K concentrations. Potassium is essential for osmotic regulation and stress tolerance, and its availability may have contributed to the higher ear density in treated plots. Likewise, elevated molybdenum in both soil and leaf tissues points to enhanced nitrate metabolism—consistent with increased soil nitrate levels—potentially driven by microbially mediated stimulation of nitrification processes. These shifts indicate improved nutrient cycling and utilization within the plant–soil system.

Together, these nutrient dynamics offer a mechanistic explanation for the agronomic improvements observed. By fostering microbial activity and facilitating nutrient transformations, CFMI-8 enhanced both soil fertility and crop performance, supporting its potential as a sustainable soil amendment.

### 4.4. Soil Health, Microbial Activity, and Crop Productivity

Figure 3 highlights key relationships between soil health, microbial activity, and productivity outcomes. Notably, the strong correlation between corn yield and corn ear density supports the idea that yield improvements may be driven by changes in both soil conditions and plant reproductive efficiency [73].

Correlations among soil organic carbon, microbial organic carbon, and overall soil health scores emphasize the importance of organic matter inputs in supporting a biologically active soil ecosystem. Such conditions foster microbial growth and function, which in turn promote nutrient cycling and structural integrity—key factors in sustaining crop yields [74]. This suggests that treatments aimed at increasing organic matter inputs (e.g., compost, microbial inoculants) could improve both microbial activity and soil structure, ultimately leading to higher yields.

Among nutrients, phosphorus and zinc emerge as potentially limiting elements for biomass accumulation. Positive associations between soil P and silage yield, and between soil Zn and plant performance, suggest that optimizing micronutrient availability—through microbial mobilization or targeted supplementation—could enhance productivity in systems where these nutrients are marginal [75,76,77].

The elevated microbial activity observed in treated soils (as indicated by respiration and microbial carbon metrics) further supports the hypothesis that microbial-mediated transformations underpin improved nutrient availability and soil function [78,79]. These observations align with treatment-induced shifts in yield components and nutrient concentrations, reinforcing the value of microbial amendments.

The treatment effects suggest that enhancing soil microbial interactions can improve nutrient cycling and crop performance [80,81]. This is evidenced by the higher soil respiration and microbial organic carbon levels in treated plots, which indicate a more active microbial community likely contributing to better soil structure and nutrient availability [82,83].

Taken together, the data suggest that interventions aimed at strengthening microbial processes and organic matter dynamics can contribute to more efficient nutrient cycling and improved crop productivity. Future studies should explore causal pathways in greater detail, ideally through multi-season trials and comparative microbial profiling [84].

### 4.5. Soil Health and Fertility

Figure 3 illustrates how CFMI-8 treatment supports soil fertility through improved microbial function and moisture dynamics. Strong correlations among Soil Health Score, Microbial Organic Carbon, and CO_2_ Respiration point to enhanced microbial biomass and activity as key drivers of soil functionality [85].

The positive association between Soil Organic Carbon and Microbial Organic Carbon reinforces the importance of organic matter in sustaining microbial communities and facilitating nutrient cycling [86]. Notably, soil moisture-related parameters also correlated with soil health outcomes: higher field capacity was linked to elevated Soil Health Scores, while the inverse relationship between Wilting Point and Soil Organic Carbon suggests that organic-rich soils retain water more effectively and create more favorable conditions for microbial processes [87,88,89].

Yield data further support the role of microbial-driven improvements in soil quality. Corn and silage yields were positively associated with Soil Health Score, and grain yield correlated strongly with Total Nitrogen—consistent with nitrogen’s role in driving crop productivity [6,89,90,91].

Taken together, these results indicate that CFMI-8 enhances microbial and moisture-related soil functions, which are integral to nutrient availability and plant performance. Future studies should evaluate how these changes persist over time and under varying environmental conditions to assess long-term sustainability impacts.

### 4.6. Random Forest Analysis and Predictive Metrics

Random Forest analysis identified microbial activity metrics (e.g., CO_2_ soil respiration, microbial organic carbon) and nitrogen availability (e.g., nitrate-N) as the strongest predictors of corn yield (Figure 4) and influential factors in micronutrient uptake (Figure 5). These findings align with previous studies emphasizing the importance of microbial processes and nitrogen dynamics in soil fertility and crop productivity [92,93,94].

### 4.7. Key Biogeochemical Predictors of Corn Productivity

As shown in Figure 4, nitrate (NO_3_-N) emerged as the strongest predictor of corn yield, indicating that nitrogen availability plays a critical role in biomass accumulation and grain development. This aligns with previous studies showing that nitrogen is a primary limiting factor in crop production [75,95].

Microbial Organic Carbon also showed high predictive power, reinforcing the importance of soil microbial activity in nutrient cycling. This finding is consistent with previous reports that increased microbial biomass enhances organic matter decomposition and nutrient availability, contributing to higher crop productivity [92,96]. Additionally, CO_2_ soil respiration was a strong predictor, highlighting microbial-driven carbon cycling as a key factor for soil health and fertility [97].

Other influential predictors include Soil Organic Carbon (LOI), Soil pH, and Soil Health Score, which collectively indicate that nutrient retention, microbial balance, and soil structure are major drivers of agronomic performance. These results support previous findings that maintaining high organic matter and microbial diversity enhances nutrient availability and plant growth [46,98].

### 4.8. Biological and Agronomic Drivers of Micronutrient Uptake

Random Forest regression revealed that both agronomic management and crop performance play key roles in micronutrient assimilation (Figure 5). Notably, treatment status and corn ear density consistently emerged as top predictors, suggesting that managed interventions and optimal plant architecture significantly enhance nutrient uptake through improved plant vigor and root development.

Yield components such as grain and silage yield were also strong predictors of leaf micronutrient concentrations. This correlation indicates that more productive plants likely develop extensive root systems, facilitating more efficient nutrient absorption and translocation.

Soil quality indicators—including soil organic carbon and soil health score—were moderately important, underscoring the role of soil organic matter in supporting microbial activity and nutrient cycling. Although soil pH, field capacity, and wilting point were included, their lower importance suggests a more indirect influence on micronutrient uptake relative to management and productivity-related variables.

For leaf-level uptake specifically, CO_2_ soil respiration was the most influential predictor (Figure 5), reinforcing the role of active microbial communities in nutrient solubilization and rhizosphere bioavailability [98]. Microbial Organic Carbon was also a strong contributor, supporting the hypothesis that soil microbial interactions are central to micronutrient exchange and plant-microbe symbiosis [44]. While the limited sample size constrains broader generalizability, these findings offer valuable preliminary insights into the link between biological soil health and nutrient uptake efficiency. Further validation in larger, multi-site datasets is warranted.

### 4.9. Limitations of the Study

While this study provides strong evidence for the positive effects of CFMI-8 on soil health and crop productivity, several limitations should be noted. First, the trial was conducted at a single site under specific environmental and soil conditions, which may limit the generalizability of the findings to other regions or soil types. Future studies across diverse agroecological zones would help confirm the broader applicability of these results. Second, although microbial activity was inferred from CO_2_ respiration and microbial organic carbon measurements, direct characterization of microbial community shifts was not performed. The absence of 16S rRNA or metagenomic sequencing precludes confirmation of specific taxa changes or microbial diversity improvements. Incorporating high-resolution sequencing approaches in future work will provide more mechanistic insight into how microbial inoculants influence soil microbiomes. Lastly, environmental variability, including fluctuations in temperature, moisture, and field microclimates, may have influenced nutrient dynamics and plant performance. Controlled multi-season studies are recommended to further validate the robustness of the observed treatment effects. Despite these limitations, this work provides a strong foundation for future research and highlights the potential of microbial inoculants as sustainable tools for improving soil function and crop yield.

While this study did not include microbiome sequencing, a parallel trial using the same microbial inoculant in a cotton cropping system has generated 16S rRNA data that confirms shifts in soil microbial community composition. These data, to be published separately, will help contextualize the functional responses observed here and provide further insight into the microbial mechanisms underpinning soil health improvements.

## 5. Conclusions

This study provides evidence that CFMI-8, a co-fermented microbial inoculant, significantly enhances soil health, nutrient cycling, and crop productivity in a field setting. Treated soils exhibited elevated microbial organic carbon (+224.1%) and CO_2_ soil respiration (+167.1%), indicating stimulated microbial activity that supports improved nutrient mineralization and organic matter turnover.

These microbial-driven changes were associated with greater nutrient availability—particularly nitrate nitrogen (+20.2%)—and improved soil fertility metrics such as cation exchange capacity (+23.1%). The result was a measurable impact on crop performance: corn yield increased by 28.6% and silage yield by 9.6% in the treated plots compared to the untreated control.

Leaf micronutrient concentrations also improved, especially for Fe, Mn, B, Zn, and Mo, suggesting enhanced uptake efficiency likely mediated by microbial activity and soil health improvements. A statistically significant 8.1% increase in leaf phosphorus, despite only a modest (non-significant) rise in soil phosphorus, further supports this interpretation.

Random forest modeling identified nitrate nitrogen, microbial organic carbon, and CO_2_ respiration as the most influential predictors of yield, reinforcing the importance of microbial processes in agricultural productivity.

While the study is limited by its single-season design, absence of strain-level tracking, and lack of economic analysis, the findings underscore CFMI-8’s promise as a regenerative input for sustainable agriculture. Future work should examine long-term effects, mechanistic interactions between microbial consortia and native soil microbiomes, and economic feasibility across diverse agronomic systems.

## Figures and Tables

**Figure 1 microorganisms-13-01638-f001:**
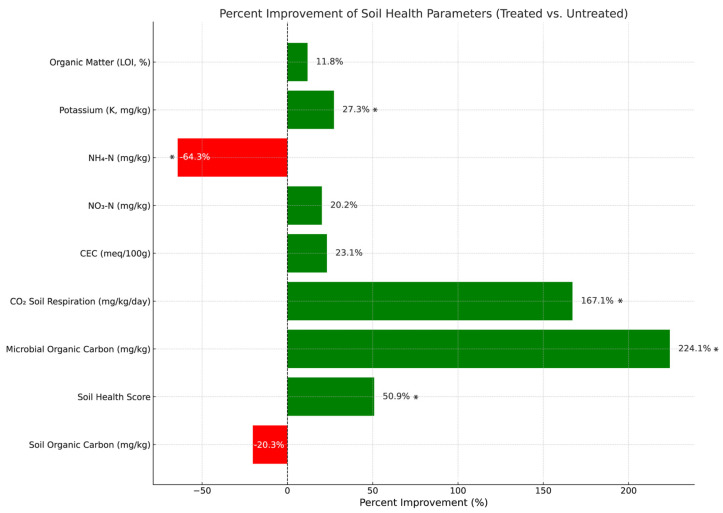
Percent change in soil health parameters (treated vs. untreated). Significant differences between treated and untreated soils (*p* < 0.05) are marked with an asterisk (*). The bar chart illustrates the percentage changes in key soil health parameters, comparing treated and untreated soils. Positive changes (green bars) indicate improvements in microbial organic carbon, soil respiration, cation exchange capacity (CEC), and nutrient availability, while negative changes (red bars) represent declines in certain parameters, such as soil organic carbon turnover. The bar chart was generated using Python, v3.12.3, utilizing libraries such as Matplotlib v.3.8.4 for visualization. The graphical representation highlights the relative impact of treatment on soil health metrics, providing a clear comparison of improvements driven by microbial inoculation.

**Figure 2 microorganisms-13-01638-f002:**
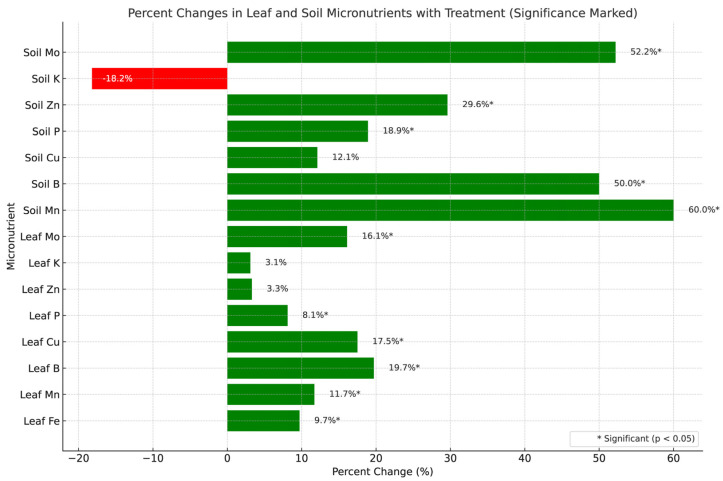
Percent changes in leaf and soil micronutrient levels following CFMI-8 treatment. The figure illustrates relative changes in micronutrient levels in corn leaves between treated and untreated groups. Green bars represent improved nutrient availability or plant uptake, while red bars indicate potential decreases, likely due to higher plant absorption. Statistically significant differences (*p* < 0.05) are marked with an asterisk (*). This visualization, generated using Python’s Matplotlib v3.8.4 [28] highlights the treatment’s impact on soil fertility and plant nutrition. Percentage changes were calculated as (Treated − Untreated)/Untreated × 100, with a horizontal bar chart distinguishing positive (green) and negative (red) shifts. The chart includes annotated values, a 0% reference line, and clear axis labels for readability.

**Figure 3 microorganisms-13-01638-f003:**
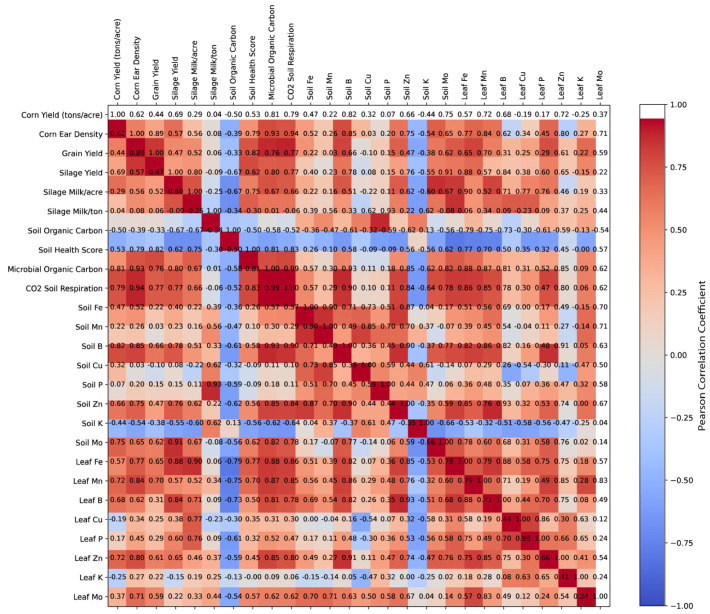
Pearson correlation matrix of soil, leaf, and productivity indicators in response to CFMI-8 treatment. This heatmap displays the Pearson correlation coefficients among key soil health metrics, leaf micronutrient content, and crop productivity outcomes. Strong positive correlations (red) indicate tightly linked variables, while negative correlations (blue) suggest inverse relationships. Notable patterns include strong associations between microbial indicators (e.g., microbial organic carbon, CO_2_ respiration) and corn yield, as well as between soil and leaf micronutrient levels. The color scale on the right reflects the strength and direction of each correlation. The heatmap was generated using Python’s Matplotlib v3.8.4and Seaborn 0.13.2 libraries.

**Figure 4 microorganisms-13-01638-f004:**
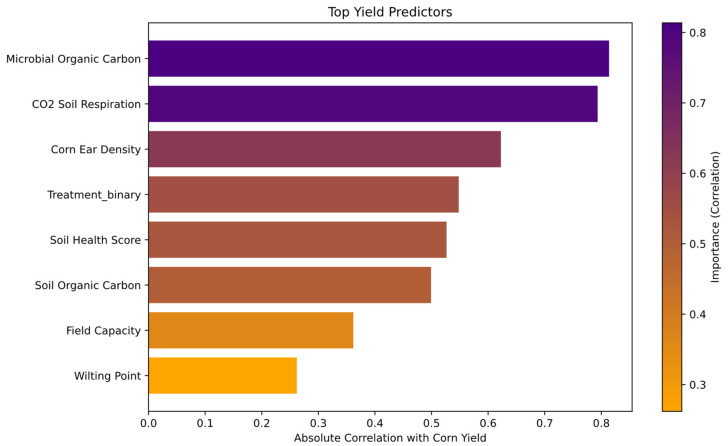
Top predictors for corn yield based on Random Forest analysis. This figure presents the most influential soil health and nutrient factors affecting corn yield, as identified through Random Forest analysis. The feature importance scores indicate the relative contribution of each factor, with Microbial Organic Carbon and CO_2_ Soil Respiration, followed Corn Ear Density and Treatment with soil inoculant were the strongest predictors. This figure was created using Random Forest feature importance analysis, where the contribution of each soil health and nutrient factor to corn yield prediction was quantified and visualized as a bar plot. Bar plot was generated using Python, v3.12.3 specifically utilizing the Scikit-Learn library v1.4.2 [26] for Random Forest modeling and permutation feature importance analysis, along with Matplotlib v3.8.4 [28] and Seaborn v0.13.2 [27] for visualization.

**Figure 5 microorganisms-13-01638-f005:**
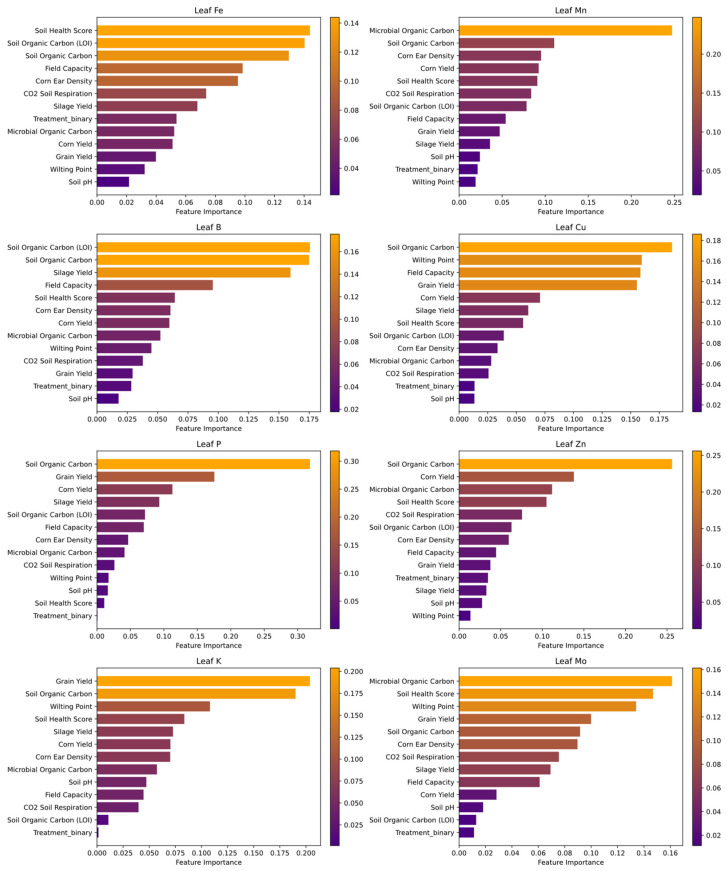
Random Forest feature importances for predicting leaf micronutrient uptake. This composite figure presents the ranked predictor importances for eight leaf micronutrients (Fe, Mn, B, Cu, P, Zn, K, and Mo), based on Random Forest models trained using soil parameters, yield components, and treatment status. Each subplot displays normalized importance scores for variables contributing to a given micronutrient, with darker bars (purple) indicating stronger predictive power. Soil Organic Carbon consistently emerged as a key predictor, particularly for Fe, B, Cu, P, and Zn. Although the limited sample size (*n* = 8) restricts statistical certainty, the results highlight potential drivers of nutrient uptake. Bar plots were generated using Scikit-Learn’s permutation importance method and visualized with Matplotlib v3.8.4 [28] and Seaborn v0.13.2 [27]. A summary of top-ranked predictors is illustrated in Figure 3.

**Table 1 microorganisms-13-01638-t001:** Comparison of soil health parameters between treated and untreated groups.

Parameter	Treated (Mean ± SD)	Untreated (Mean ± SD)
Soil Organic Carbon (mg/kg)	118.00 ± 5.62	148.00 ± 11.37
Soil Health Score	13.26 ± 2.33	8.79 ± 0.80
Microbial Organic Carbon (mg/kg)	80.48 ± 23.54	24.83 ± 5.18
CO_2_ Soil Respiration (mg/kg/day)	95.83 ± 30.34	35.88 ± 5.92
Cation Exchange Capacity (meq/100 g)	8.78 ± 0.22	7.13 ± 0.55
NO_3_-N (mg/kg)	38.15 ± 3.58	31.75 ± 4.43
NH_4_-N (mg/kg)	0.6 ± 0.08	1.68 ± 0.41
Potassium (K, mg/kg)	368.75 ± 11.14	289.75 ± 34.11
Organic Matter (LOI, %)	1.90 ± 0.07	1.70 ± 0.07

**Table 2 microorganisms-13-01638-t002:** Summary of agronomic performance metrics under CFMI-8 and control treatments.

Parameter	Treated (Mean ± SD)	Untreated (Mean ± SD)	% Change
Corn Yield (tons/acre)	7.20 ± 1.79	5.60 ± 0.68	+28.6%
Corn Ear Density	38.50 ± 1.87	32.75 ± 2.05	+17.6%
Grain Yield	246.40 ± 6.68	235.57 ± 7.03	+4.6%
Silage Yield (tons/acre)	36.26 ± 1.18	33.08 ± 1.51	+9.6%
Silage Milk/acre (lbs.)	121,703.00 ± 6320.59	111,333.75 ± 3652.58	+9.3%
Silage Milk/ton (lbs.)	3426.00 ± 183.84	3332.25 ± 67.36	+2.8%

**Table 3 microorganisms-13-01638-t003:** Comparison of means, standard deviations, percent changes, and *p*-values between treated and untreated groups across soil variables and corn leaf nutrient content.

Metric	Untreated		Treated		% Change	*p*-Value
	Mean	Std Dev	Mean	Std Dev		
Soil Fe	31.500	7.188	65.750	28.745	108.730	0.094
Soil Mn	1.250	0.500	2.000	0.816	60.000	0.178
Soil B	0.300	0.000	0.450	0.058	50.000	0.014
Soil Cu	0.575	0.150	0.650	0.238	13.043	0.617
Soil P	56.750	4.500	67.500	12.583	18.943	0.188
Soil Zn	1.350	0.058	1.750	0.129	29.630	0.004
Soil K	130.000	25.245	106.250	36.087	−18.269	0.327
Soil Mo	0.023	0.005	0.035	0.010	55.556	0.083
Leaf Fe	110.500	2.380	121.250	2.754	9.729	0.001
Leaf Mn	31.100	1.738	34.750	0.957	11.736	0.016
Leaf B	10.150	0.819	12.150	0.473	19.704	0.009
Leaf Cu	10.000	0.816	11.750	2.217	17.500	0.216
Leaf P	0.372	0.015	0.402	0.017	8.054	0.039
Leaf Zn	2.128	0.025	2.203	0.025	3.525	0.005
Leaf K	32.750	1.708	33.750	1.708	3.053	0.439
Leaf Mo	0.310	0.036	0.355	0.013	14.516	0.080

## Data Availability

All data generated or analyzed during this study are included in this published article. Additional information is available from the corresponding author upon reasonable request.

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
