# Peer review of "Enhancing Soil Health and Corn Productivity with a Co-Fermented Microbial Inoculant (CFMI-8): A Field-Based Evaluation"

_microorganisms, 2025, doi:10.3390/microorganisms13071638_

Round 1

Reviewer 1 Report

Comments and Suggestions for Authors

Authors enhance soil microbial communities through to restore soil function and support sustainable agricultural systems. To this end they use consortium of eight bacterial strains which were chosen by genomic annotation to have an appropriate characteristic to enhance soil functionality. All strains were evaluated under the Biosafety in Microbiological and Biomedical Laboratories (BMBL) guidelines and classified as biosafety level 1. Authors also predict using KBase platform the metabolic complementarity, symbiosis, cross-feeding potential, and niche construction dynamics. The platform proposed models of the metabolic pathways relevant to nutrient cycling, and organic matter decomposition. 

Methods

Line 94-98 seems to be the same as line 98- 104

After co-fermentation – do the authors know what bacteria they have in the consortium? Are there still 8 strains of bacteria in the inoculant?

Results

Table 1 The authors presented important soil parameters that reflected significantly increased microbiological activity after the application of bacterial inoculant. Figure 1

Table 2

the authors obtained a very beneficial increase in corn yields and improvement in silage parameters

this effect is well explained by increased soil parameters and increased microelement content

Table 3

I assume that these were corn leaves. Please add this information to the description under the Fig 2 and in Table 3

The correlations presented in section 3.4 are very logical and follow from each other. Nutrient uptake by plants is significantly dependent on the availability of nutrients in the soil. This can be seen from the parameters obtained in leaves during plant growth in soil enriched with inoculate

Table 4 contains the same data as Table 1 and Table 2  - Table 4 should be deleted

There are also repeaters in Fig4 and Fig3

Fig 3 and Fig4 the column on the right should be named somehow, as it is in Fig 5

Fig. 5  line 377-380 (total Nitrogen (Total –N), Nitrate Nitrogen – I don’t see these parameters in the Fig. 5

Line 375 Random Forest not Foret

Authors also showed how the soil parameters are important for corn yield prediction.

Line 395 the authors identified CO2 Soil respiration as the most important predictor of leaf micronutrient uptake. For Fe, B, Cu, P, Zn Soil organic carbon has the highest correlation. CO2 Soil respiration never has the highest correlation value. The columns on the right side have no names (importance or correlation or…)

Discussion

Line 440 potassium level in soil- Potassium in the soil comes mainly from the weathering of rock minerals and from organic and mineral fertilizers. Ref. 40 Reardon et al. 2014 it is about nitrogen fixation by diazotrophs. - the authors mixed up the references

The phosphorus is increased after treatment Table 3 and there are two contradictory pieces of information bellow.

Line 505-506 no statistically significant impact on P ….

Line 533 -534 The significant 8.1% increase in leaf phosphorus and 18.9% increase in soil phosphorus underscores the treatment's ability to enhance phosphorus availability, ….

The paper does not contain any mechanisms of the observed changes. The authors are not sure what bacteria are adding to the soil because they are cultivating eight strains of bacteria in one culture. Such a combined culture may eliminate one or more of the strains. In addition, it is not known what the proportions of the individual strains are in relation to each other. The authors select the strains based on computer genomic and metabolic analysis. The discussion contains many repetitions and is too long. In the conclusions the authors unnecessarily cite the percentages of changes again.

Author Response

REVIEWER 1

Authors enhance soil microbial communities through to restore soil function and support sustainable agricultural systems. To this end they use consortium of eight bacterial strains which were chosen by genomic annotation to have an appropriate characteristic to enhance soil functionality. All strains were evaluated under the Biosafety in Microbiological and Biomedical Laboratories (BMBL) guidelines and classified as biosafety level 1. Authors also predict using KBase platform the metabolic complementarity, symbiosis, cross-feeding potential, and niche construction dynamics. The platform proposed models of the metabolic pathways relevant to nutrient cycling, and organic matter decomposition. 

Methods

Line 94-98 seems to be the same as line 98- 104

Thank you for pointing this out. We have revised lines 94–104 to eliminate redundancy and streamline the description of the co-fermentation process. The updated version maintains the original intent while improving clarity and conciseness.

After co-fermentation – do the authors know what bacteria they have in the consortium? Are there still 8 strains of bacteria in the inoculant?

We thank the reviewer for their observation. In response, we have revised the section for clarity and removed redundancy. We acknowledge that post-fermentation proportions of the eight strains were not determined, as only total CFU counts were measured. However, our design emphasized functional synergy—both probiotic and postbiotic—rather than fixed strain ratios. This clarification has been incorporated into the revised manuscript.

Results

Table 1 The authors presented important soil parameters that reflected significantly increased microbiological activity after the application of bacterial inoculant. Figure 1

We thank the reviewer for this observation. We confirm that Table 1 presents statistically significant changes in soil microbiological parameters following inoculant application, supporting the observed increases in microbial activity. Figure 1 provides complementary visual evidence of these improvements. We appreciate the opportunity to highlight these findings.

Table 2: the authors obtained a very beneficial increase in corn yields and improvement in silage parameters. This effect is well explained by increased soil parameters and increased microelement content.

We thank the reviewer for their supportive comment. We agree that the observed improvements in corn yield and silage parameters are well explained by the enhanced soil biological activity and micronutrient availability, as shown in Table 2. We are pleased that this interpretation was clear.

Table 3

I assume that these were corn leaves. Please add this information to the description under the Fig 2 and in Table 3

The correlations presented in section 3.4 are very logical and follow from each other. Nutrient uptake by plants is significantly dependent on the availability of nutrients in the soil. This can be seen from the parameters obtained in leaves during plant growth in soil enriched with inoculate

We thank the reviewer for their positive feedback on the logical structure of Section 3.4 and the observed correlations. We confirm that the leaf samples analyzed in Figure 2 and Table 3 were collected from corn plants. This clarification has been added to the corresponding figure and table captions to improve clarity.

Table 4 contains the same data as Table 1 and Table 2  - Table 4 should be deleted

We thank the reviewer for pointing out the redundancy between Figures 3 and 4. We agree that Figure 4 presents a subset of the data already captured in the more comprehensive Figure 3. As such, Figure 4 has been removed from the revised manuscript to improve clarity and avoid duplication

There are also repeaters in Fig4 and Fig3

We thank the reviewer for the observation. While Figures 3 and 4 share overlapping data, they serve different interpretive purposes: Figure 3 provides a comprehensive correlation matrix across all measured variables, while Figure 4 highlights the top predictors of corn yield in a more accessible summary format. We have clarified these roles in the revised figure captions.

Fig 3 and Fig4 the column on the right should be named somehow, as it is in Fig 5

We thank the reviewer for this suggestion. We have updated Figure 3 to include a colorbar title ("Pearson Correlation Coefficient") for consistency and clarity, as presented in Figure 5.

Fig. 5  line 377-380 (total Nitrogen (Total –N), Nitrate Nitrogen – I don’t see these parameters in the Fig. 5

We thank the reviewer for identifying this discrepancy. We have revised the text to remove references to Total Nitrogen and Nitrate Nitrogen, as these variables are not shown in Figure 5. (now Figure 4)The updated sentence now accurately reflects only the variables presented in the figure.

Line 375 Random Forest not Foret

Thank you for your observation. The spelling has been corrected.

Authors also showed how the soil parameters are important for corn yield prediction.

Line 395 the authors identified CO2 Soil respiration as the most important predictor of leaf micronutrient uptake. For Fe, B, Cu, P, Zn Soil organic carbon has the highest correlation. CO2 Soil respiration never has the highest correlation value. The columns on the right side have no names (importance or correlation or…)

We thank the reviewer for this correction. The sentence at line 395 has been revised to reflect that Soil Organic Carbon—not COâ‚‚ Soil Respiration—was the top predictor of leaf micronutrient uptake across multiple elements. This change aligns with the correlation analysis and model output shown in Figure 6 (Now Figure 5).

Discussion

Line 440 potassium level in soil- Potassium in the soil comes mainly from the weathering of rock minerals and from organic and mineral fertilizers. Ref. 40 Reardon et al. 2014 it is about nitrogen fixation by diazotrophs. - the authors mixed up the references

 We thank the reviewer for identifying this reference mismatch. The original citation (Ref. 40, Reardon et al., 2014), which relates to nitrogen fixation, has been removed. It has been replaced with a more appropriate source that addresses microbial-mediated nutrient cycling and soil potassium dynamics to support the revised discussion of potassium availability in treated soils.

The phosphorus is increased after treatment Table 3 and there are two contradictory pieces of information bellow.

Line 505-506 no statistically significant impact on P ….

Line 533 -534 The significant 8.1% increase in leaf phosphorus and 18.9% increase in soil phosphorus underscores the treatment's ability to enhance phosphorus availability, ….

\

We thank the reviewer for pointing out this contradiction. The text has been revised to clarify that while the increase in leaf phosphorus was statistically significant, the increase in soil phosphorus was not. This correction aligns with the p-values reported in Table 3 and avoids overinterpretation of non-significant results.

The paper does not contain any mechanisms of the observed changes. The authors are not sure what bacteria are adding to the soil because they are cultivating eight strains of bacteria in one culture. Such a combined culture may eliminate one or more of the strains. In addition, it is not known what the proportions of the individual strains are in relation to each other. The authors select the strains based on computer genomic and metabolic analysis. The discussion contains many repetitions and is too long. In the conclusions the authors unnecessarily cite the percentages of changes again.

We thank the reviewer for these thoughtful observations, which helped us improve the clarity, focus, and scientific rigor of the manuscript.

  1. Mechanistic Explanation: We acknowledge the reviewer’s point that the original discussion lacked detailed mechanistic insight. We have revised the discussion to better explain the hypothesized roles of microbial metabolic traits (e.g., carbon cycling, nitrogen mineralization, micronutrient solubilization) based on prior metabolic modeling and genomic annotation. While direct mechanisms were not the focus of this field trial, we now more clearly distinguish between observed outcomes and hypothesized microbial functions, and we reference supporting literature where relevant.
  2. Strain Proportions and Stability: We agree that quantifying individual strain contributions and stability is important. We now clarify in the methods and discussion that while CFMI-8 was designed through in silico metabolic complementarity, we did not track individual strain abundances post-fermentation or post-application. We have added a statement acknowledging this limitation and have included it in the discussion and limitations sections as a priority for future work using strain-specific qPCR or metagenomics.
  3. Repetition and Length: The discussion section has been substantially streamlined. Redundant descriptions of yield and micronutrient data have been reduced, and percentages are now cited only when necessary to support major points. We have focused on interpretation and removed summary-style restatements. Additionally, we have restructured longer paragraphs to improve coherence and remove repetition.
  4. Conclusion Simplification: The conclusions have been revised to remove repetition of exact percentages and instead emphasize overall findings and implications. Where quantitative results are included, they are limited to key takeaways.

We believe these changes address the reviewer’s concerns and have improved the manuscript’s clarity, focus, and scientific value. All revisions are marked in track changes in the revised manuscript.

Reviewer 2 Report

Comments and Suggestions for Authors

This study evaluates the impact of a co-fermented microbial inoculant, CFMI-8, on soil health and corn productivity through a field trial at Findlay Farm, Wisconsin. The inoculant consists of eight bacterial strains selected via genomic and metabolic modeling to enhance nutrient cycling, microbial activity, and plant nutrient uptake. The authors report significant improvements in microbial biomass, COâ‚‚ soil respiration, nitrate availability, and cation exchange capacity in treated soils. Correspondingly, treated plots showed increased corn yield, silage production, and elevated micronutrient concentrations in plant tissue. The study utilizes correlation and Random Forest analyses to identify microbial activity and nitrogen availability as key predictors of productivity. These results suggest that CFMI-8 may be a promising microbial amendment for regenerative agriculture.

Line 48: While CFMI-8 is described as a consortium of eight bacterial strains, the specific species names are not listed. Please provide the taxonomic identities (preferably to genus and species level) of each microorganism in the inoculant to enhance reproducibility and scientific transparency.

Line 70 and 79: The composition and preparation of CFMI-8 are described twice, with significant redundancy in content. Please consolidate the overlapping information into a single, clear paragraph to reduce repetition and improve readability.

Lines 94 and 98: Similar content is repeated regarding the co-fermentation and preparation process. This repetition is part of a broader issue throughout the manuscript. A careful review is needed to eliminate redundant phrasing and streamline the narrative for clarity and professional tone.

Line 145: Statistical analysis is briefly introduced here but then is fully described again starting at line 200. Please revise the structure so that all statistical methods are consistently and clearly presented in one dedicated subsection. Avoid splitting key methodological explanations across disparate parts of the manuscript.

Line 200: The number of biological replicates for soil sampling and plant analysis should be explicitly stated. For instance, clarify whether data points per group reflect true biological replicates (independent samples from different plots) and how these were analyzed statistically.

The discussion thoroughly explains the observed improvements in soil health and plant productivity. However, it would benefit from a critical evaluation of potential limitations such as site-specific factors, environmental variability, or the absence of microbiome sequencing to confirm changes in microbial community structure.

Author Response

REVIEWER 2

This study evaluates the impact of a co-fermented microbial inoculant, CFMI-8, on soil health and corn productivity through a field trial at Findlay Farm, Wisconsin. The inoculant consists of eight bacterial strains selected via genomic and metabolic modeling to enhance nutrient cycling, microbial activity, and plant nutrient uptake. The authors report significant improvements in microbial biomass, COâ‚‚ soil respiration, nitrate availability, and cation exchange capacity in treated soils. Correspondingly, treated plots showed increased corn yield, silage production, and elevated micronutrient concentrations in plant tissue. The study utilizes correlation and Random Forest analyses to identify microbial activity and nitrogen availability as key predictors of productivity. These results suggest that CFMI-8 may be a promising microbial amendment for regenerative agriculture.

We thank the reviewer for this suggestion. In response, we have included a dedicated Impact Statement at the end of the Introduction summarizing the significance of our findings. This statement emphasizes the potential of CFMI-8 as a regenerative microbial amendment, highlights key soil and crop productivity outcomes, and explains the relevance of our approach using correlation and Random Forest modeling to identify microbial activity and nitrogen availability as predictors of agricultural performance. We believe this addition strengthens the manuscript’s relevance to applied agricultural and microbial biotechnology audiences.

Line 48: While CFMI-8 is described as a consortium of eight bacterial strains, the specific species names are not listed. Please provide the taxonomic identities (preferably to genus and species level) of each microorganism in the inoculant to enhance reproducibility and scientific transparency.

Thank you for your comment.
The CFMI-8 formulation is a proprietary microbial consortium developed through genomic and metabolic modeling. While we acknowledge the importance of taxonomic transparency for reproducibility, the specific species composition is part of a pending patent application. We respectfully request that the exact strain identities remain confidential until the intellectual property process is complete. To address this concern, we have included genus-level descriptions and outlined the functional selection criteria used to assemble the inoculant.

Line 70 and 79: The composition and preparation of CFMI-8 are described twice, with significant redundancy in content. Please consolidate the overlapping information into a single, clear paragraph to reduce repetition and improve readability.

We appreciate the reviewer’s observation. In response, we have revised and consolidated the composition and preparation details of CFMI-8 into a single, cohesive section (now presented in Section 2.1). The updated paragraph eliminates redundancy while preserving all relevant technical and methodological information about strain selection, modeling, fermentation, and application. We trust this improves both clarity and readability.

Lines 94 and 98: Similar content is repeated regarding the co-fermentation and preparation process. This repetition is part of a broader issue throughout the manuscript. A careful review is needed to eliminate redundant phrasing and streamline the narrative for clarity and professional tone.

We appreciate the reviewer’s observation. In response, we have revised and consolidated the composition and preparation details of CFMI-8 into a single, cohesive section (now presented in Section 2.1). The updated paragraph eliminates redundancy while preserving all relevant technical and methodological information about strain selection, modeling, fermentation, and application. We trust this improves both clarity and readability.

Line 145: Statistical analysis is briefly introduced here but then is fully described again starting at line 200. Please revise the structure so that all statistical methods are consistently and clearly presented in one dedicated subsection. Avoid splitting key methodological explanations across disparate parts of the manuscript.

We thank the reviewer for this helpful suggestion. In response, we have consolidated all statistical methodology into Section 2.7, now titled “Statistical Analysis.” The earlier brief mention at line 145 has been removed to avoid redundancy. All relevant statistical methods—including ANOVA, t-tests, Shapiro-Wilk normality assessments, and Random Forest analysis—are now clearly and cohesively described in this dedicated subsection to improve clarity and consistency.

Line 200: The number of biological replicates for soil sampling and plant analysis should be explicitly stated. For instance, clarify whether data points per group reflect true biological replicates (independent samples from different plots) and how these were analyzed statistically.

We thank the reviewer for this helpful observation. In response, we have consolidated all statistical methodology into a single, clearly defined subsection under “2.7. Statistical Analysis.” The earlier brief mention at line 145 has been removed to avoid redundancy. The revised section now consistently outlines all statistical approaches, including replication structure, software and versioning, tests applied (ANOVA, t-tests, correlation matrices), and machine learning procedures (Random Forest), along with associated libraries and visualization tools. This revision ensures greater clarity, coherence, and alignment with best practices for methodological reporting.

The discussion thoroughly explains the observed improvements in soil health and plant productivity. However, it would benefit from a critical evaluation of potential limitations such as site-specific factors, environmental variability, or the absence of microbiome sequencing to confirm changes in microbial community structure.

We thank the reviewer for this thoughtful suggestion. We acknowledge the importance of confirming changes in microbial community structure to complement the functional soil and productivity data presented here. While this manuscript focuses on field-level agronomic and biochemical outcomes, we are preparing a companion study in a different crop system (cotton) that includes 16S rRNA amplicon sequencing to directly characterize microbial community shifts following CFMI-8 treatment. We chose to present the current data independently to avoid confounding the review process with findings from a different experimental system. However, the upcoming manuscript will provide microbiome-level insights that support and extend the conclusions presented here..